# Neural Network Modeling of Microstructure Formation in an AlMg6/10% SiC Metal Matrix Composite and Identification of Its Softening Mechanisms under High-Temperature Deformation

Alexander Smirnov * , Vladislav Kanakin and Anatoly Konovalov

Institute of Engineering Science, UB RAS, 34 Komsomolskaya St., 620049 Ekaterinburg, Russia
* Correspondence: smirnov@imach.uran.ru

**Abstract:** The paper investigates the rheological behavior and microstructuring of an AlMg6/10% SiC metal matrix composite (MMC). The rheological behavior and microstructuring of the AlMg6/10% SiC composite is studied for strain rates ranging between 0.1 and 4 s$^{-1}$ and temperatures ranging from 300 to 500 °C. The microstructure formation is studied using EBSD analysis, as well as finite element simulation and neural network models. The paper proposes a new method of adding data to a training sample, which allows neural networks to correctly predict the behavior of microstructure parameters, such as the average grain diameter, and the fraction and density of low-angle boundaries with scanty initial experimental data. The use of neural networks has made it possible to relate the thermomechanical parameters of deformation to the microstructure parameters formed under these conditions. These dependences allow us to establish that, at strain rates ranging from 0.1 to 4 s$^{-1}$ and temperatures between 300 to 500 °C, the main softening processes in the AlMg6/10% SiC MMC are dynamic recovery and continuous dynamic recrystallization accompanied, under certain strain and strain rate conditions at 300 and 350 °C, by geometric recrystallization.

**Keywords:** metal matrix composite; high temperature; aluminum; simulation; rheology; neural network; relaxation; rheological behavior; flow stress; Al-Mg alloy; Al-Mg-Sc-Zr alloy

## 1. Introduction

Aluminum and aluminum alloys have become widely used in industry due to their high specific mechanical properties, high thermal conductivity, and corrosion resistance, as well as other important technological parameters. Numerous aluminum-based alloys doped with various alloying elements offering the required physical and mechanical properties of an alloy have been created. These materials include Al-Mg alloys, which have good corrosion resistance, ductility, and weldability [1–6]. The latter alloys are used as structural materials for cryogenic structures, and they are used in the aerospace industry. Magnesium additives in aluminum strengthen it significantly. Each percent of Mg content increases the tensile strength by 25–30 MPa [7]. At the same time, Al-Mg alloys can be considered as deformable with a Mg content of up to 11–12 wt%. With a Mg content of up to 8 wt%, these alloys cannot be hardened by heat treatment [7]. However, with a magnesium content of more than 7 wt%, its anticorrosive properties deteriorate sharply; therefore, Al-Mg alloys with this Mg content are hardly ever used. This leads to the fact that Al-Mg alloys used in industry are not amenable to heat treatment, and they have medium strength. As a result, these alloys find limited use only in some areas of the industry.

One of the ways to increase the strength properties of Al-Mg alloys is their alloying with the Sc rare earth element, which, even in a small content, increases the strength of alloys [8,9]. The hardening of Al-Mg alloys by adding scandium is a combined effect of dispersion hardening and structural hardening [10,11]. These alloys are generally alloyed

with Sc in an amount of up to 0.3% and modified with Mn and Zr [11–14]. However, despite the low content of scandium due to its high cost, the use of Al-Mg-Sc alloys is also limited.

Another way to obtain increased material properties, which has recently gained increasing attention from the industry and researchers, is the creation of metal matrix composites based on an aluminum alloy matrix reinforced with various fillers in various percentages [15–20]. The development of a new material consists not only in synthesis but also in the need to form the required properties after its manufacture. The required properties of alloys and alloy-based composites, as well as products made of them, are formed using heat treatment and machining in a wide temperature range. In particular, for structural materials, these types of processing make it possible to obtain materials and products with compromised plastic and strength properties. Thus, it becomes possible to control the properties using controlled thermomechanical action consisting of mechanical action on the workpiece under certain temperature conditions [21–23]. As a rule, the thermomechanical processing of alloys and alloy-based composites is carried out at elevated temperatures, at which the alloy structure is actively rearranged.

During plastic deformation, these materials undergo competing processes associated with hardening due to the increased dislocation density and phase transformations, as well as with softening due to dynamic recrystallization, recovery, and increased damage [23–32]. The interaction of hardening and softening affects the rheological behavior of the material. In particular, the flow stress curves at certain temperatures and strain rates may have a peak [26,33–35], a steady-state portion [25–27,34,35], and several hardening and softening sections [10,11,36]; in addition, there may be an inverse strain rate dependence [37–39], when an increase in strain rate causes a decrease in flow stress. To determine the conditions for the formation of the required shape of the workpiece from a material with the necessary properties, one is to relate the thermomechanical parameters of deformation to flow stress and the microstructure parameters. These relationships are generally established by means of mathematical models based on the functional relation of microstructure parameters to thermomechanical parameters [40–43], as well as using physically based models [35,44–49] and computational models, including models based on the cellular automata method [34,50–54] and molecular dynamics [55]. Neural network models for describing microstructure formation under high-temperature deformation have not found wide application due to the need to obtain a large amount of experimental data which is time-consuming and leads to the inexpediency of using neural networks. The problem of the scanty experimental data is solved by simulating microstructure formation using cellular automata or embedding functions into finite element programs [56–59]. Simulation enables computational experiments to be conducted, thus making it possible to reduce the time spent on forming a sample of the necessary size for training neural networks. Nevertheless, this approach is rather time-consuming as it requires first developing or applying a model that adequately describes the microstructure formation in the material and then using the results of calculations based on the model to train neural networks. This paper proposes a new and relatively simple method for processing scarce experimental data relating microstructure parameters to the parameters of thermomechanical action on a metal material for subsequent neural network training. The neural networks constructed in this study are used to identify the softening mechanisms in an AlMg6/10% SiC metal matrix composite at temperatures from 300 to 500 °C and strain rates ranging between 0.1 and $4 \, \text{s}^{-1}$, as well as to relate the microstructure parameters of the metal matrix composite to the thermomechanical parameters of deformation.

## 2. Materials and Methods

### 2.1. Material and Research Technique

The AlMg6/10% SiC metal matrix composite (MMC) [60] produced using powder technology is used as the material for the study. The initial components of the AlMg6/10% SiC MMC were mixed in a vibratory mixer in an argon atmosphere. Sintering was carried

out for 60 min at a temperature of 420 °C. The pressure at which sintering occurred was 30 MPa. No additives modifying the surface of SiC particles were used.

The size of the SiC particles corresponds to the F1500 standard, with an average diameter of 2.0 ± 0.4 µm. Figure 1 shows an EBSD image and SEM image of the composite before deformation. As can be seen from this figure, the matrix particles are a polycrystalline material. After sintering of the composite, the SiC reinforcer particles are located along the boundaries of the matrix particles. In this paper, the composite was studied without pre-extrusion. Its mechanical properties at room temperature are shown in Table 1. For comparison, this table shows the properties of AlMg6 and Al-Mg-Sc-Zr alloys, which are similar in use to the composite.

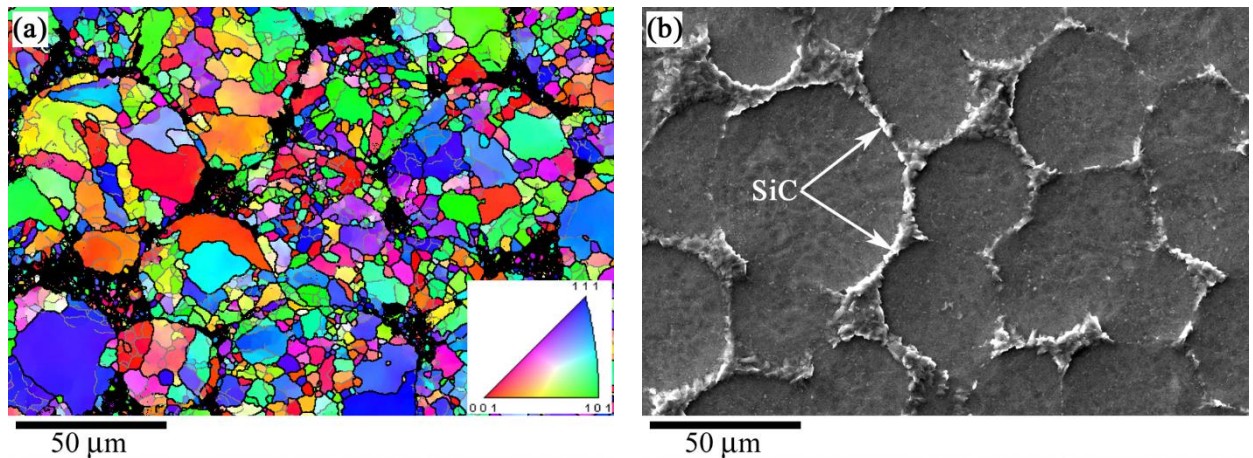

**Figure 1.** (**a**) The EBSD image after electropolishing and (**b**) the SEM image of the AlMg6/10% SiC MMC microstructure.

**Table 1.** Material properties.

| Material | Yield Strength, MPa | Elastic Modulus, GPa | Density, g/cm$^3$ |
|---|---|---|---|
| AlMg6/10%SiC | 179 | 93 | 2.67 |
| Al-Mg-Sc-Zr alloy | 164 | 66 | 2.65 |
| AlMg6 alloy | 141 | 76 | 2.62 |

Before testing, all the materials were identically annealed for 15 h at 520 °C. It can be seen from this table that the yield strength and elastic modulus of the composite are higher than those of the AlMg6 alloy and its more expensive Al-Mg-Sc-Zr analog. Table 2 shows the chemical composition of the Al-Mg-Sc-Zr alloy, the AlMg6 alloy, and the matrix of the AlMg6/10% SiC metal matrix composite. The chemical compositions were determined by means of a Spectromaxx LMF04 optical emission spectrometer, and the granulometric composition of the powders was determined by laser diffraction using a LaSca-TD.

**Table 2.** Chemical composition of materials, wt%.

| Material | Al | Mg | Mn | Sc | Fe | Zr | Si | Cu | Ti | Zn | Be |
|---|---|---|---|---|---|---|---|---|---|---|---|
| AlMg6 alloy and the matrix of AlMg6/10% SiC | balance | 6.56 | 0.5 | - | 0.27 | - | 0.16 | 0.013 | 0.04 | 0.02 | 0.0012 |
| Al-Mg-Sc-Zr alloy | balance | 5.18 | 0.36 | 0.23 | 0.12 | 0.07 | 0.01 | 0.022 | 0.02 | 0.02 | 0.003 |

The densities of the materials were determined using the hydrostatic method according to ASTM B311-13, by weighing the specimens in air and distilled water on an Ohaus Pioneer PA 214 analytical balance.

Experiments on the determination of the mechanical properties and rheological behavior of the composite at room and high temperatures were performed for cylindrical specimens. Compression specimens had a diameter $d_0$ of $6 \pm 0.05$ mm and a height $h_0$ of $9 \pm 0.05$ mm. Cylindrical tensile specimens had the following dimensions: $d_0 = 5 \pm 0.05$ mm, and the length of the gauge part $l_0 = 25 \pm 0.1$ mm. The data obtained from high-temperature compression tests of specimens were used to construct flow stress curves depending on temperature, strain, and strain rate. These curves were used for finite element simulation of specimen compression. Compression experiments were carried out using an automated plastometric installation designed at the Institute of Engineering Science, UB RAS [11,39].

In compression experiments, a graphite-containing lubricant was used to reduce friction between the punch and specimen. The lubricant provided the coefficient of Coulomb friction between the flat die and aluminum alloy equal to 0.09 at 300 °C and equal to 0.1 and 0.13 at 400 and 500 °C, respectively. The values of the friction coefficients were obtained using the procedure described in [61], the essence of which is to compress specimens to different heights and select the friction coefficient in the Coulomb friction law according to the results of finite element simulation in such a way that the maximum and minimum diameters of the specimen coincide as closely as possible with the simulated one at different strains.

In the temperature range between 300 and 500 °C, the specimens became barrel-shaped despite the use of the lubricant (Figure 2). The specimens were cooled immediately after the end of deformation, and in 2 s, the specimen temperature did not exceed 70 °C. After deformation, the specimens were cut in a longitudinal section (parallel to the compression axis) using electric spark cutting. Then, thin sections for EBSD analysis were made in the plane of the longitudinal section of the specimens. Zones of the specimens for EBSD analysis are highlighted in orange in Figure 2. For the same zones, the strain $\varepsilon$ and strain rate $\dot{\varepsilon}$ were calculated on the basis of finite element simulation of compression of specimens at the studied range of temperatures, strains and strain rates. The formulation of the finite element simulation problem is given in Section 2.2.

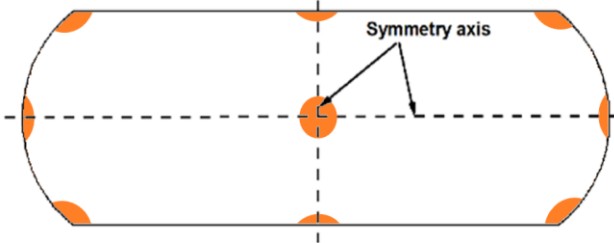

**Figure 2.** The longitudinal section of a cylindrical specimen after compression. The zones of the EBSD analysis of the specimen microstructure are highlighted in orange.

The microstructures of the specimens before and after deformation were studied by electron backscattered diffraction (EBSD) using a Tescan Vega II scanning electron microscope with an Oxford HKL Nordlys F+ EBSD analysis accessory. For the EBSD analysis, the specimens were first mechanically polished and then electropolished by 90% $CH_3COOH$ +10% $HClO_4$ electrolyte cooled to 8 °C. The polishing time averaged 6 s at a voltage of 40 V and a current density of 0.3 A/mm$^2$. The scanning step during the EBSD analysis was equal to 0.3 μm. The size of the scanned area was $300 \times 150$ μm. It was assumed that the grain misorientation exceeded 15° and that the subgrain misorientation ranged between 2 and 15°.

### 2.2. Formulation of the Finite Element Problem for Specimen Compression

In order to determine the stress–strain state in the specimen, its finite element model was constructed in the Deform program. For the specimen material, an isothermal viscoplastic model of isotropic strain hardening was used. The flow stress was set in a tabular

form based on cylindrical specimen compression tests (Figure 3a). The calculations were performed under the assumption of the axisymmetric stress–strain state and deformation symmetry about the horizontal geometric symmetry axis of the specimen (Figure 2). Thus, only a quarter of the specimen cross-section was simulated (Figure 3b).

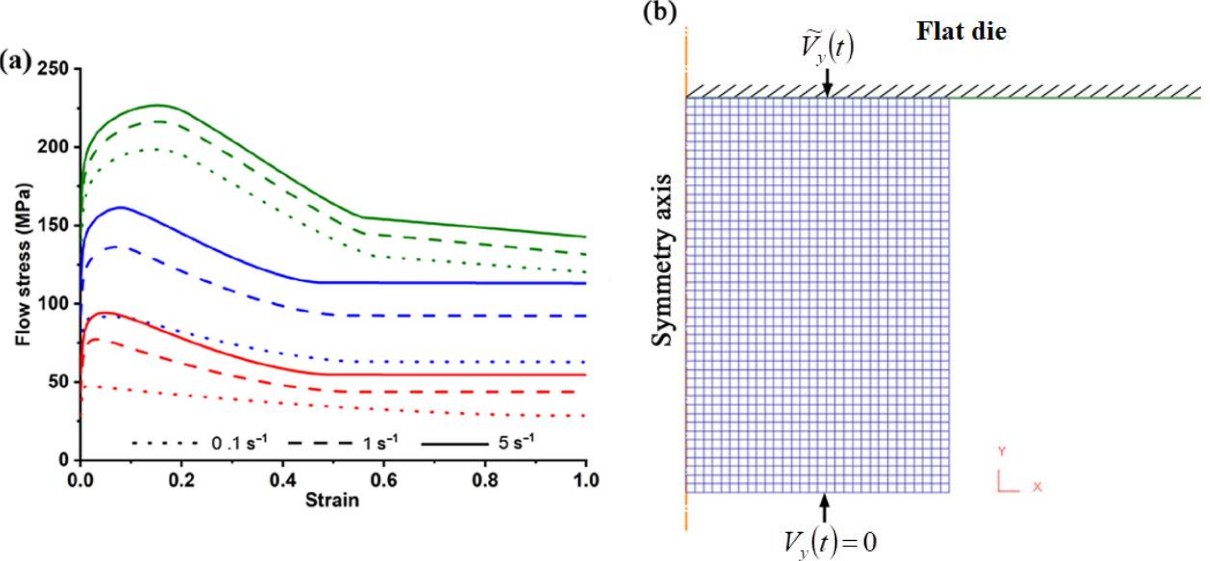

**Figure 3.** (**a**) Flow stress as dependent on strain for 300 °C (green curve), 400 °C (blue curve), 500 °C (red curve), and strain rates ranging between 0.1 and 10 s$^{-1}$; (**b**) the finite element grid used for the simulation.

The simulation was performed in accordance with the experimental conditions of specimen loading in which the time dependence of die speed $\widetilde{V}_y(t)$ (Figure 3b) was taken for each specific tested specimen. It was assumed that the friction between the specimen and the die followed Coulomb's friction law and depended on temperature. In the simulation of specimen deformation, the Coulomb friction coefficient was set to 0.09 at a temperature of 300 °C and to 0.1 and 0.13 at 400 and 500 °C, respectively.

### 2.3. Neural Network Models for Forming Microstructure Parameters

The average grain diameter $D$, the fraction of low-angle boundaries $P_L$, and their density $S_L$ were the analyzed parameters of the composite matrix microstructure depending on temperature, strain, and strain rate. The fraction and density of low-angle boundaries were calculated using the formulas

$$P_L = \frac{L}{L+H} \text{ and } S_L = \frac{L}{F} \tag{1}$$

Here, $L$ is the sum of the lengths of all the low-angle boundaries; $H$ is the sum of the lengths of all the high-angle boundaries; and $F$ is the area of the microstructure image on which the sum of the lengths of all the low-angle boundaries $L$ was determined. These parameters were determined from the data obtained by EBSD. The average grain diameter $D$ was calculated based on a sample that included grains containing at least 3 indexed points. To describe the evolution of the microstructure parameters, we used one-, two-, and three-layer neural networks. Their general scheme is shown in Figure 4. The neural networks were built using the scikit-learn library.

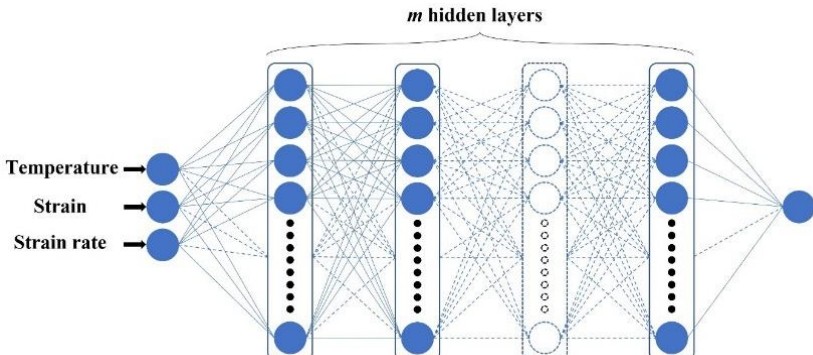

**Figure 4.** General scheme of the neural network model.

### 3. Results

*3.1. The Rheological Behavior and Microstructuring of the Composite*

Under high deformation temperatures, relaxation processes occur in metallic materials and composites based on them, which affect the form of flow stress curves. The flow stress curves for the AlMg6/10% SiC MMC (Figure 3a) can be divided into several portions (stages). At stage I, the composite undergoes hardening until the peak (maximum) flow stress value is reached. Moreover, the strain corresponding to the peak stress decreases with increasing temperature and increases with an increasing strain rate. The hardening stage is followed by a portion corresponding to material softening, where the flow stress value decreases with increasing strain. At stage III (the steady-state portion), the hardening and softening rates are close, and the flow stress value remains almost unchanged with increasing strain. This rheological behavior of the composite at high temperatures means that dynamic recrystallization occurred during deformation, which can follow the discontinuous, continuous, or geometric recrystallization mechanisms [23]. In order to identify the recrystallization mechanism, an EBSD analysis of the specimens was performed after various thermomechanical loading conditions.

Figures 5–7 show the EBSD images of the microstructures of the AlMg6/10% SiC MMC after deformation as dependent on temperature, strain, and strain rate. The black spots in the figures correspond to non-indexed zones, which generally are zones with a high content of silicon carbide falling out from the matrix during electrochemical polishing. The values of strain $\varepsilon$ and the strain rate $\dot{\varepsilon}$ were determined from the results of the finite element simulation of specimen compression. The formulation of the finite element problem is described in Section 2.

Although the specimens have the same height after compression, the accumulated strain in the same zones varies at different temperatures (Figure 8). Thus, it is difficult to analyze the occurring softening processes depending on temperature and strain-rate loading conditions without using any approximating equations. Table 3 shows the experimentally obtained values of the average grain diameter $D$, and the fraction $P_L$ and density $S_L$ of low-angle boundaries depending on temperature, strain, and strain rate. According to these data, maps of the formation of the average grain diameter, and the fraction and density of low-angle boundaries are plotted against the thermomechanical parameters of deformation in Figures 9–11. As can be seen from these maps, increasing strain decreases the average grain diameter, while an increasing deformation temperature forms larger grains in the composite matrix. The effect of strain rate on grain evolution is nonmonotonic. In the grain map (Figure 9), there are portions where the diameter of the formed grains decreases with an increasing strain rate, and there are portions where, in contrast, the average grain diameter decreases with a decreasing strain rate.

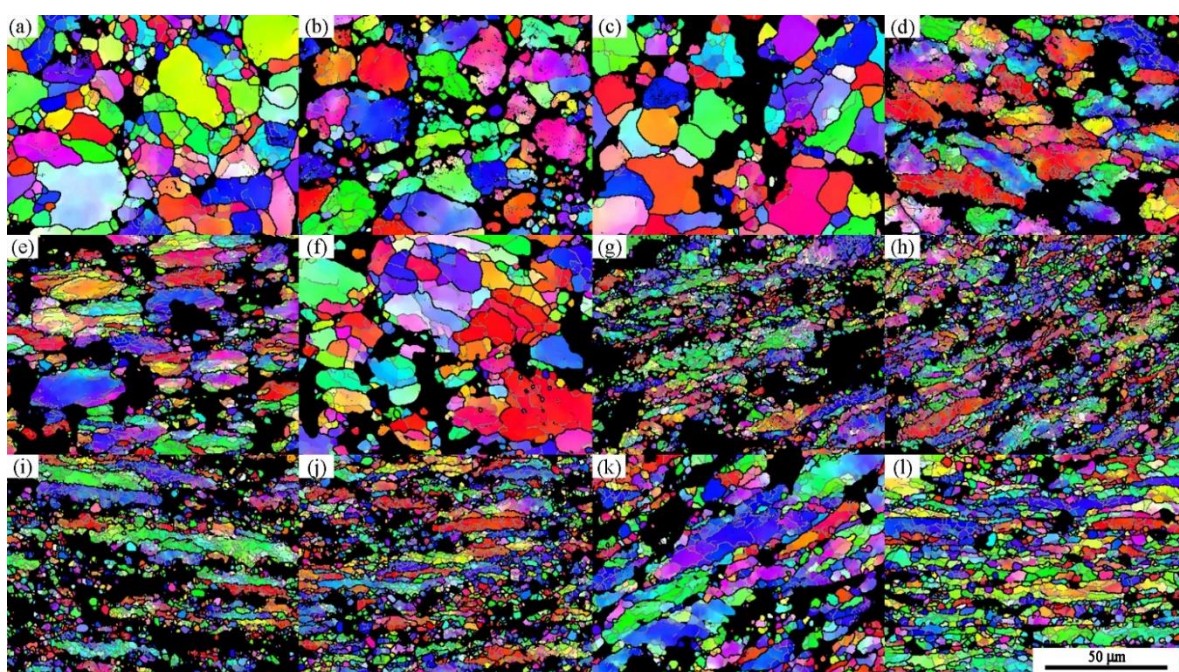

**Figure 5.** An EBSD image of the composite matrix microstructure at a deformation temperature of 300 °C for (**a**) $\varepsilon = 0.12$ and $\dot{\varepsilon} = 0.83$ s$^{-1}$; (**b**) $\varepsilon = 0.13$ and $\dot{\varepsilon} = 0.26$ s$^{-1}$; (**c**) $\varepsilon = 0.17$ and $\dot{\varepsilon} = 0.05$ s$^{-1}$; (**d**) $\varepsilon = 0.37$ and $\dot{\varepsilon} = 2.48$ s$^{-1}$; (**e**) $\varepsilon = 0.40$ and $\dot{\varepsilon} = 0.8$ s$^{-1}$; (**f**) $\varepsilon = 0.45$ and $\dot{\varepsilon} = 0.15$ s$^{-1}$; (**g**) $\varepsilon = 0.58$ and $\dot{\varepsilon} = 3.85$ s$^{-1}$; (**h**) $\varepsilon = 0.63$ and $\dot{\varepsilon} = 1.25$ s$^{-1}$; (**i**) $\varepsilon = 0.63$ and $\dot{\varepsilon} = 4.18$ s$^{-1}$; (**j**) $\varepsilon = 0.66$ and $\dot{\varepsilon} = 1.32$ s$^{-1}$; (**k**) $\varepsilon = 0.73$ and $\dot{\varepsilon} = 0.24$ s$^{-1}$; (**l**) $\varepsilon = 0.76$ and $\dot{\varepsilon} = 0.26$ s$^{-1}$.

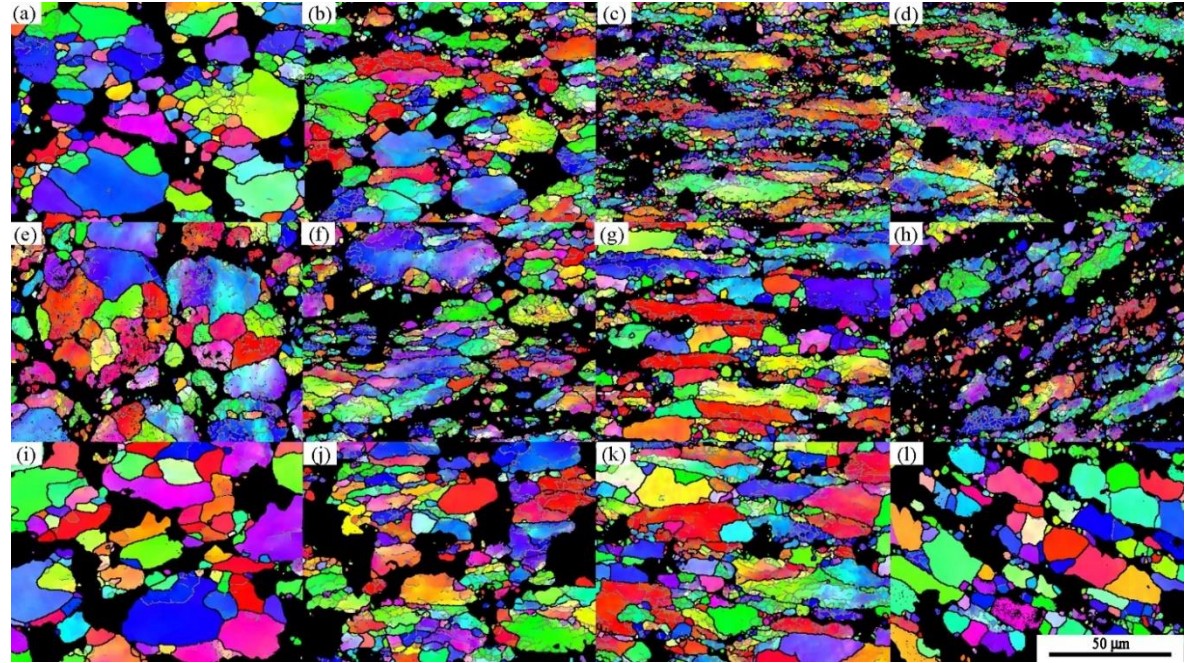

**Figure 6.** An EBSD image of the composite matrix microstructure at a deformation temperature of 400 °C for (**a**) $\varepsilon = 0.07$ and $\dot{\varepsilon} = 0.54$ s$^{-1}$; (**b**) $\varepsilon = 0.11$ and $\dot{\varepsilon} = 0.22$ s$^{-1}$; (**c**) $\varepsilon = 0.17$ and $\dot{\varepsilon} = 0.05$ s$^{-1}$; (**d**) $\varepsilon = 0.34$ and $\dot{\varepsilon} = 2.32$ s$^{-1}$; (**e**) $\varepsilon = 0.43$ and $\dot{\varepsilon} = 0.81$ s$^{-1}$; (**f**) $\varepsilon = 0.49$ and $\dot{\varepsilon} = 0.15$ s$^{-1}$; (**g**) $\varepsilon = 0.60$ and $\dot{\varepsilon} = 4.06$ s$^{-1}$; (**h**) $\varepsilon = 0.63$ and $\dot{\varepsilon} = 4.29$ s$^{-1}$; (**i**) $\varepsilon = 0.79$ and $\dot{\varepsilon} = 1.47$ s$^{-1}$; (**j**) $\varepsilon = 0.83$ and $\dot{\varepsilon} = 0.25$ s$^{-1}$; (**k**) $\varepsilon = 0.88$ and $\dot{\varepsilon} = 1.64$ s$^{-1}$; (**l**) $\varepsilon = 0.92$ and $\dot{\varepsilon} = 0.28$ s$^{-1}$.

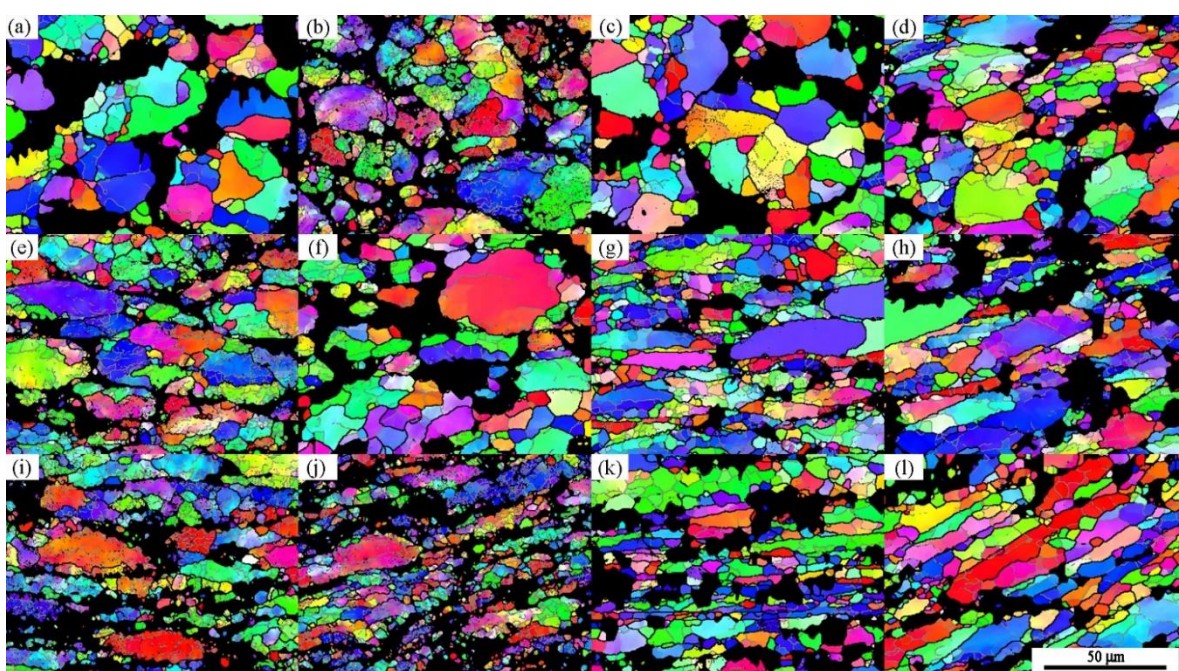

**Figure 7.** An EBSD image of the composite matrix microstructure at a deformation temperature of 500 °C for (**a**) $\varepsilon = 0.02$ and $\dot{\varepsilon} = 0.3\,\mathrm{s}^{-1}$; (**b**) $\varepsilon = 0.05$ and $\dot{\varepsilon} = 0.1\,\mathrm{s}^{-1}$; (**c**) $\varepsilon = 0.1$ and $\dot{\varepsilon} = 0.03\,\mathrm{s}^{-1}$; (**d**) $\varepsilon = 0.33$ and $\dot{\varepsilon} = 2.36\,\mathrm{s}^{-1}$; (**e**) $\varepsilon = 0.4$ and $\dot{\varepsilon} = 0.81\,\mathrm{s}^{-1}$; (**f**) $\varepsilon = 0.51$ and $\dot{\varepsilon} = 0.16\,\mathrm{s}^{-1}$; (**g**) $\varepsilon = 0.66$ and $\dot{\varepsilon} = 4.57\,\mathrm{s}^{-1}$; (**h**) $\varepsilon = 0.73$ and $\dot{\varepsilon} = 5.0\,\mathrm{s}^{-1}$; (**i**) $\varepsilon = 0.77$ and $\dot{\varepsilon} = 1.54\,\mathrm{s}^{-1}$; (**j**) $\varepsilon = 0.92$ and $\dot{\varepsilon} = 1.84\,\mathrm{s}^{-1}$; (**k**) $\varepsilon = 0.97$ and $\dot{\varepsilon} = 0.29\,\mathrm{s}^{-1}$; (**l**) $\varepsilon = 1.15$ and $\dot{\varepsilon} = 0.35\,\mathrm{s}^{-1}$.

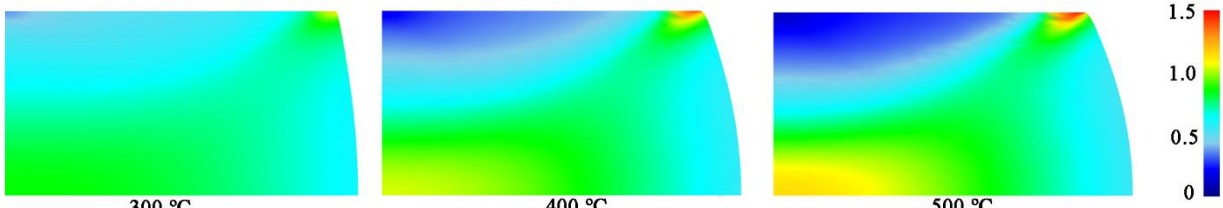

**Figure 8.** Accumulated strain distribution in the AlMg6/10% SiC MMC specimens under 50% compression at 300, 400, and 500 °C.

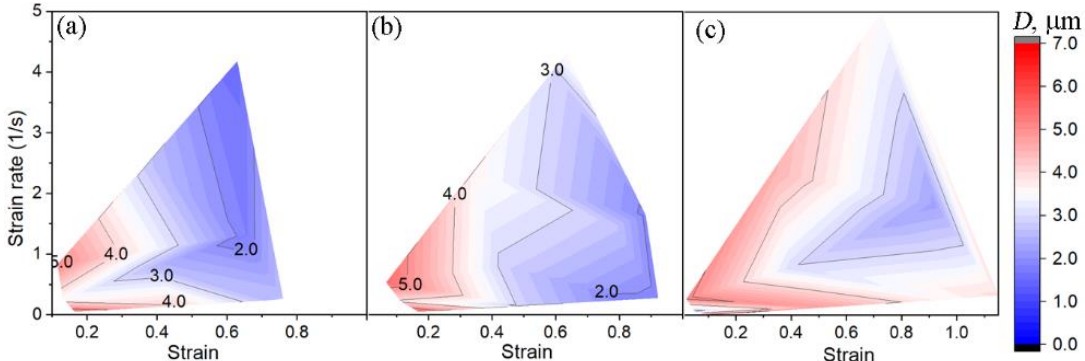

**Figure 9.** The effect of strain and strain rate on the average grain diameter depending on temperature, °C: (**a**) 300, (**b**) 400, (**c**) 500. The dependences are plotted from the results of the EBSD analysis of the microstructure.

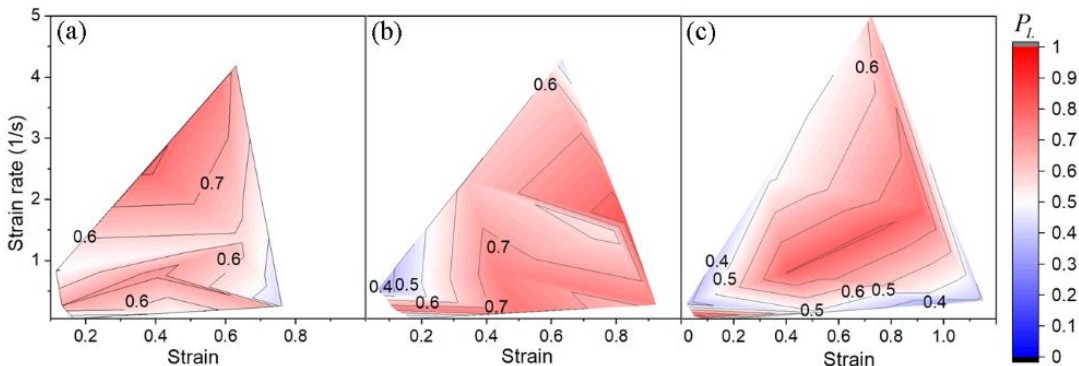

**Figure 10.** The effect of strain and strain rate on the fraction of low-angle boundaries depending on temperature, °C: (**a**) 300, (**b**) 400, (**c**) 500. The dependences are plotted from the results of the EBSD analysis of the microstructure.

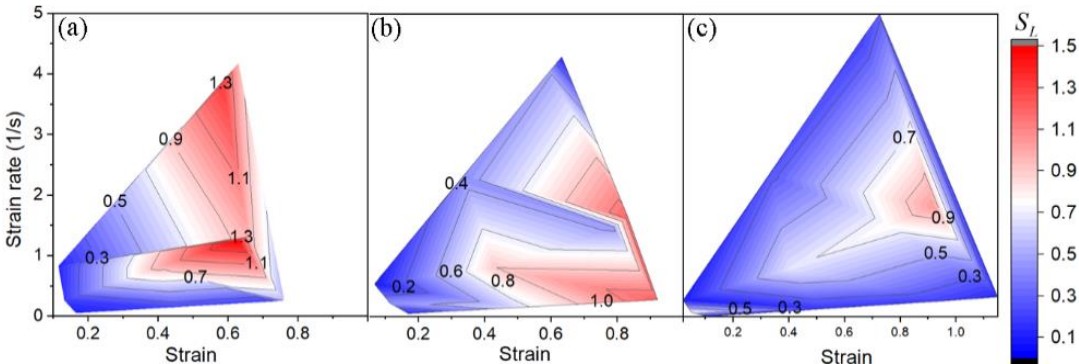

**Figure 11.** The effect of strain and strain rate on the density of low-angle boundaries depending on temperature, °C: (**a**) 300, (**b**) 400, (**c**) 500. The dependences are plotted from the results of the EBSD analysis of the microstructure.

**Table 3.** Microstructure parameters of the AlMg6/10 % SiC MMC at temperatures of 300, 400, and 500 °C.

| Temp., °C | | | | | | 300 | | | | | |
|---|---|---|---|---|---|---|---|---|---|---|---|
| Strain | 0.12 | 0.13 | 0.17 | 0.37 | 0.40 | 0.45 | 0.58 | 0.63 | 0.63 | 0.66 | 0.73 | 0.76 |
| Strain rate, $s^{-1}$ | 0.83 | 0.26 | 0.05 | 2.48 | 0.8 | 0.15 | 3.85 | 1.25 | 4.18 | 1.32 | 0.24 | 0.26 |
| D, µm | 5.07 | 3.52 | 5.45 | 2.73 | 2.56 | 4.49 | 1.67 | 1.82 | 1.61 | 1.83 | 3.33 | 2.68 |
| $P_L$ | 0.5 | 0.7 | 0.43 | 0.82 | 0.73 | 0.55 | 0.76 | 0.64 | 0.67 | 0.57 | 0.58 | 0.43 |
| $S_L$, $µm^{-1}$ | 0.13 | 0.17 | 0.1 | 0.73 | 0.97 | 0.23 | 1.30 | 1.50 | 0.97 | 0.97 | 0.20 | 0.53 |

| Temp., °C | | | | | | 400 | | | | | |
|---|---|---|---|---|---|---|---|---|---|---|---|
| Strain | 0.07 | 0.11 | 0.17 | 0.34 | 0.43 | 0.49 | 0.60 | 0.63 | 0.79 | 0.88 | 0.83 | 0.92 |
| Strain rate, $s^{-1}$ | 0.54 | 0.22 | 0.05 | 2.32 | 0.81 | 0.15 | 4.06 | 4.29 | 1.47 | 0.25 | 1.64 | 0.28 |
| D, µm | 5.46 | 3.89 | 5.6 | 3.63 | 2.9 | 2.89 | 2.98 | 3.66 | 2.73 | 1.81 | 1.92 | 1.86 |
| $P_L$ | 0.35 | 0.77 | 0.38 | 0.63 | 0.76 | 0.79 | 0.60 | 0.44 | 0.57 | 0.65 | 0.85 | 0.76 |
| $S_L$, $µm^{-1}$ | 0.07 | 0.43 | 0.13 | 0.43 | 0.9 | 0.63 | 0.47 | 0.13 | 0.47 | 1.10 | 1.20 | 1.20 |

| Temp., °C | | | | | | 500 | | | | | |
|---|---|---|---|---|---|---|---|---|---|---|---|
| Strain | 0.02 | 0.05 | 0.1 | 0.33 | 0.4 | 0.51 | 0.66 | 0.73 | 0.77 | 0.92 | 0.97 | 1.15 |
| Strain rate, $s^{-1}$ | 0.3 | 0.1 | 0.03 | 2.36 | 0.81 | 0.16 | 4.57 | 5.00 | 1.54 | 1.84 | 0.29 | 0.35 |
| D, µm | 5.13 | 2.85 | 5.27 | 4.56 | 3.04 | 4.76 | 3.37 | 3.53 | 2.43 | 2.27 | 3.55 | 3.91 |
| $P_L$ | 0.38 | 0.83 | 0.39 | 0.49 | 0.8 | 0.49 | 0.46 | 0.62 | 0.81 | 0.79 | 0.34 | 0.39 |
| $S_L$, $µm^{-1}$ | 0.10 | 0.70 | 0.10 | 0.27 | 0.67 | 0.17 | 0.20 | 0.27 | 0.77 | 1.03 | 0.20 | 0.13 |

The comparison of the initial microstructure (Figure 1a) with the microstructure formed during deformation (Figures 5–7) testifies that, after high-temperature deformation, small grains appear in the microstructure. It can also be seen that inside the initial large grains, there are subgrains with low-angle boundaries (highlighted in gray in Figures 5–7). At the same time, new elongated grains and nearly equiaxed ones are formed in the initial elongated grains along the composite matrix flow direction; the prevailing majority of them retain their geometry throughout the range of temperatures and strain rates under study. At a strain of at least 0.7, some subgrains have an almost equiaxed shape, and the other ones are elongated but with a greater equiaxiality coefficient than the grains containing these subgrains. Another experimentally determined fact is that the matrix grains are formed inside large, deformed grains. Thus, we can say that the grain formation process in the composite matrix is associated with the mechanism of continuous dynamic recrystallization [23,43]. At the same time, the subboundary migration rate in the matrix is high enough for a significant number of the subgrains to remain equiaxed. Note also that the high-angle boundaries of the new grains also have a migration rate sufficient for a significant part of the recrystallized grains to be equiaxed at strains of at least 0.7. The obtained experimental data confirm the need to construct physically based mathematical models of microstructure evolution that take into account the migration of low- and high-angle boundaries during continuous dynamic recrystallization [43,44,62,63].

It was reported in [11,62,64] that, during continuous dynamic recrystallization, the fraction of low-angle boundaries changes nonmonotonically with increasing strain. That is, the curve representing the strain dependence of the fraction of low-angle boundaries may have portions of decreasing and increasing fractions of low-angle boundaries. It is obvious from the plotted map of the formation of low-angle boundaries (Figure 10) that the composite under study has similar nonmonotonic dependencies.

It can be seen from the microstructure image (Figures 5–7) that increasing temperature induces the formation of larger subgrains and that, at a temperature of 500 °C, in some large grains, there are no subgrains at all (Figure 7). This temperature effect stems from the increasing rate of dislocation annihilation due to an increase in the velocity of their chaotic motion caused by increasing temperature [23].

It was noted above that, in real experiments at high temperatures and large plastic deformations, due to friction, the strain and strain rate loading conditions can hardly be maintained constant. As a result, at the same speed of the punch of the testing machine but different temperatures, the specimen strain rate is different. To estimate the influence of the thermomechanical parameters on microstructure formation, the experimental values of $D$, $P_L$, and $S_L$ must be determined at the same strains and strain rates; therefore, the values of the microstructure parameters obtained for different thermomechanical conditions must be approximated. In this paper, polynomials are used to plot maps of the formation of grains and low-angle boundaries (Figures 9–11) depending on temperature, strain, and strain rate. However, it is not possible to use polynomials to predict nonmonotonically changing data due to a low prediction level. As a result, Figures 9–11 show the maps of the formation of grains and low-angle boundaries only within the experimental range. For a complete analysis of the forming microstructure in the entire strain and strain rate range studied, the constructed maps are insufficient. This problem can be solved by the use of neural networks. They allow us to predict changes in the parameters with acceptable engineering accuracy if the correct architecture and training data are chosen [65,66].

*3.2. Constructing the Architecture of Neural Networks and Their Training*

To describe changes in the average grain diameter $D$, the fraction of low-angle boundaries $P_L$, and the density of low-angle boundaries $S_L$ during deformation at temperatures between 300 and 500 °C, three neural networks with a general scheme (multilayer perceptron) were constructed (Figure 4). The neural networks were trained according to the experimental data from Table 3. Verification was performed according to the experimental data from Table 4, which were obtained at temperatures and strain rates different from those

used for training. The images of the microstructures obtained with the thermomechanical deformation parameters from Table 4 are shown in Figure 12.

**Table 4.** Microstructure parameters of the AlMg6/10 % SiC MMC at temperatures of 350 and 450 °C.

| Temp., °C | 350 | | | 450 | | |
|---|---|---|---|---|---|---|
| Strain | 0.10 | 0.35 | 0.54 | 0.08 | 0.34 | 0.54 |
| Strain rate, s$^{-1}$ | 0.30 | 1.14 | 1.75 | 0.30 | 1.20 | 1.90 |
| $D$, μm | 4.41 | 3.02 | 2.19 | 4.71 | 3.58 | 2.67 |
| $P_L$ | 0.59 | 0.80 | 0.64 | 0.44 | 0.64 | 0.54 |
| $S_L$, μm$^{-1}$ | 0.06 | 0.18 | 0.23 | 0.03 | 0.15 | 0.19 |

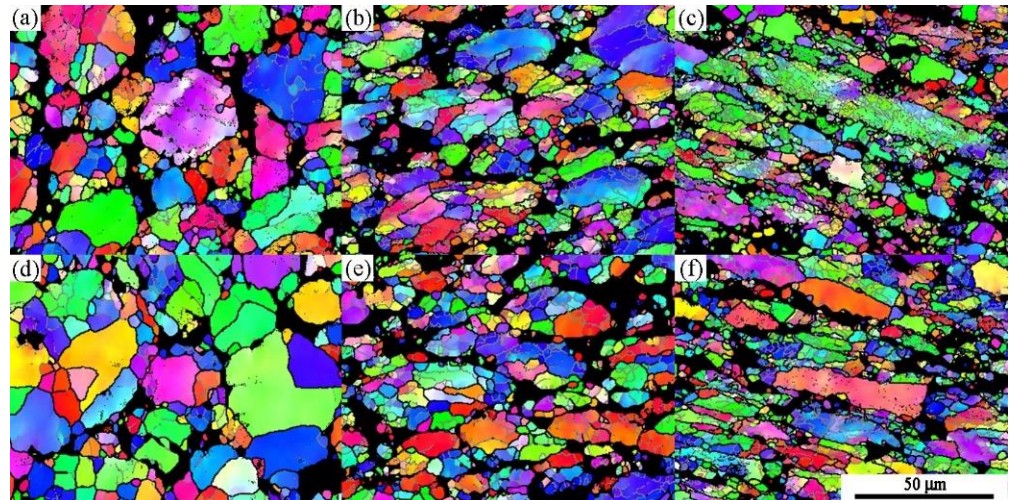

**Figure 12.** EBSD images of the composite matrix microstructure at deformation temperatures (**a**–**c**) of 350 °C (**d**–**f**) and 450 °C, which correspond to the following strain and strain rate conditions: (**a**) $\varepsilon = 0.10$ and $\dot{\varepsilon} = 0.30$ s$^{-1}$; (**b**) $\varepsilon = 0.35$ and $\dot{\varepsilon} = 1.14$ s$^{-1}$; (**c**) $\varepsilon = 0.54$ and $\dot{\varepsilon} = 1.75$ s$^{-1}$; (**d**) $\varepsilon = 0.08$ and $\dot{\varepsilon} = 0.30$ s$^{-1}$; (**e**) $\varepsilon = 0.34$ and $\dot{\varepsilon} = 1.20$ s$^{-1}$; (**f**) $\varepsilon = 0.54$ and $\dot{\varepsilon} = 1.90$ s$^{-1}$.

As it was shown in Section 3.1, the average grain diameter, and the fraction and density of low-angle boundaries vary nonmonotonically. Therefore, in order to take into account correctly the nonmonotonicity of the relations of the microstructure parameters to the thermomechanical parameters of deformation, which have a form similar to those shown in Figures 9–11, it is necessary to use a large number of neurons (usually more than 20). It should be borne in mind that the number of weight coefficients that need to be determined by training significantly exceeds the number of neurons in the hidden layer. If we choose a neural network structure with one hidden layer and three input neurons (Figure 4), the number of the weight coefficients of the neural network is four times the number of neurons in the hidden layer. In this study, 36 sets of experimental data relating the average grain diameter to temperature, strain, and strain rate were obtained. Similar sets were obtained for the fraction and density of low-angle boundaries. Thus, the maximum number of possible neurons in the hidden layer with such a single-layer neural network is nine. A neural network with this number of neurons makes it possible to describe the evolution of the average grain diameter depending on the thermomechanical parameters of deformation with an average relative deviation of 111%. The average relative deviation for each investigated microstructure parameter is calculated using the formula

$$\delta = \frac{1}{N} \sum_{i=1}^{N} \frac{|\varphi_i - z_i|}{z_i} \cdot 100\% \tag{2}$$

where $\varphi_i$ and $z_i$ are the calculated and experimental values of the compared values obtained under the same conditions (in this paper, these values are the average grain diameter $D$,

and the fraction $P_L$ and density $S_L$ of low-angle boundaries); and $N$ is the total number of compared values.

To increase the number of hidden neurons in the neural network, it is necessary to increase the size of the sample by which the neural network is trained. Augmentation is one of the possible ways of increasing the sample size by which the neural network is trained. The essence of this method is experimental data noising. Certainly, this method sometimes allows the problem of neural network uncertainty to be solved. However, as computational experiments have shown, it is impossible to achieve a significant reduction in the average relative deviation for the dependences obtained in this study. As a result, a method of processing experimental data was proposed, which consists of approximating experimental data using a surface, followed by adding data to the training sample. The values of the microstructure parameters from an arbitrary point of the constructed approximating surface are added to the training sample. Let us now consider the algorithm for creating a training sample for the average grain diameter obtained at 500 °C (Figure 13). In this figure, the experimental points are shown in red.

1. First, planes are built on the three nearest points. Then, we select a part of the plane bound by straight intersections with neighboring planes and with a triangle form (Figure 13a).
2. We remove the planes located inside the body. Here, the body means a part of the space bound by the constructed triangles.
3. If the line connecting the experimental point with its projection on the strain–strain-rate plane intersects the triangles, they are removed from the constructed surface of the body (the removed planes are shown in yellow in Figure 13a). This results in a surface approximated by planes in a certain strain and strain rate range (Figure 13b). In this case, a one-to-one correspondence of the strain and strain rate values to the microstructure parameter is achieved.
4. We take an arbitrary point of the constructed approximating surface and add its coordinates to the training sample.

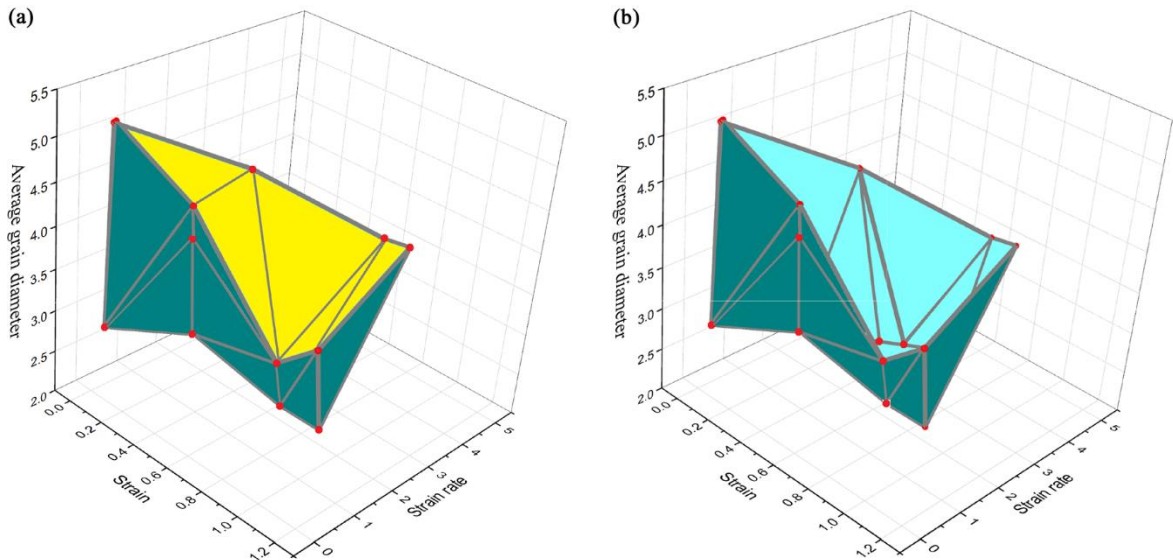

**Figure 13.** (**a**) Constructing a surface approximated by planes and (**b**) the finally constructed surface based on experimental points. The light color in (**b**) shows the planes that are hidden by the yellow planes in (**a**).

According to this technique, 1814 values for three test temperatures were included in the training sample. Similar procedures for constructing a training sample were used for experimental data describing the evolution of the fraction and density of low-angle boundaries.

To describe the evolution of the parameters of the composite matrix microstructure depending on the thermomechanical parameters of deformation and single-, two-, and three-layer networks, as well as various activation functions, were considered. The criterion for choosing the composition of the hidden layer and the activation function was the minimum value of the average relative deviation of the experimental data from the calculated ones. Based on this, a single-layer network with 100 neurons in the hidden layer was selected for the dependence of the average grain diameter $D$ and the fraction of low-angle boundaries $P_L$ on the thermomechanical parameters; a two-layer neural network with 75 neurons in the first hidden layer and 11 neurons in the second hidden layer shows the best results for the dependence of the density of the low-angle boundaries $S_L$ on the thermomechanical parameters (Figure 4). A logistic activation function was chosen for all the three neural networks as neural networks using it show the best results in approximating the microstructure parameters depending on the thermomechanical parameters of deformation. The average relative deviation of the experimental data from the calculated ones for the selected neural networks during their training was 5.4, 4.8, and 4.6% for the average grain diameter $D$, and the fraction $P_L$ and density $S_L$ of low-angle boundaries, respectively.

In order to test the ability of a neural network to correctly predict these microstructure parameters depending on the thermomechanical conditions, neural networks were verified under deformation conditions different from the conditions of training. These microstructure parameters and the corresponding thermomechanical parameters are summarized in Table 4. The average relative deviation of the experimental data from the predicted ones was 4.2, 5.8, and 5.6% for the average grain diameter $D$, and the fraction $P_L$ and density $S_L$ of low-angle boundaries, respectively. The obtained error values are acceptable for predicting the evolution of the microstructure parameters, and this enables the constructed neural networks to be used to study the mechanisms of softening in the AlMg6/10% SiC metal matrix composite. The constructed neural networks were used to obtain maps of the microstructure parameters, which describe the behavior of the average grain diameter $D$, and the fraction $P_L$ and density $S_L$ of low-angle boundaries as dependent on temperature, strain, and strain rate. These maps are shown in Figures 14–16.

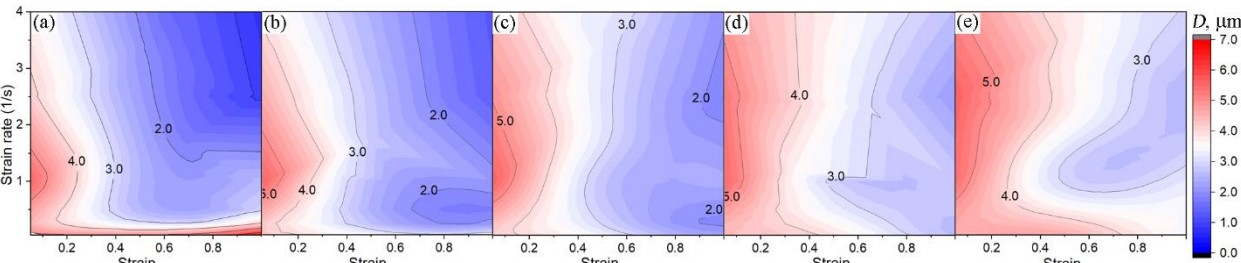

**Figure 14.** The effect of strain and strain rate on the average grain diameter $D$ depending on temperature, °C: (**a**) 300, (**b**) 350, (**c**) 400, (**d**) 450, and (**e**) 500. The dependences have been obtained from neural network predictions.

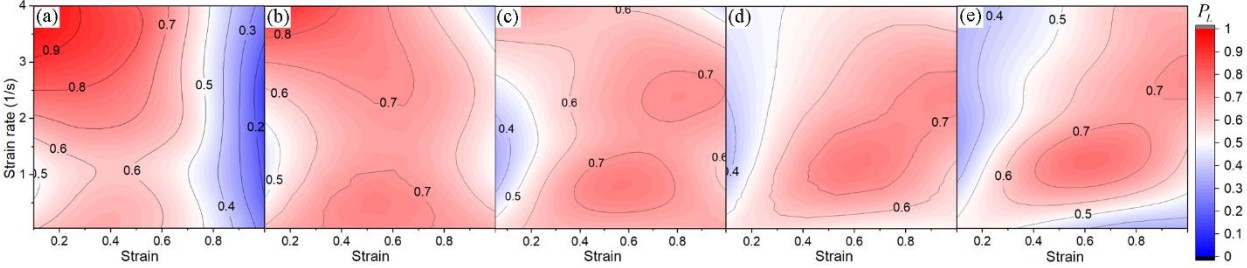

**Figure 15.** The effect of strain and strain rate on the fraction of low-angle boundaries $P_L$ depending on temperature, °C: (**a**) 300, (**b**) 350, (**c**) 400, (**d**) 450, and (**e**) 500. The dependences have been obtained from neural network predictions.

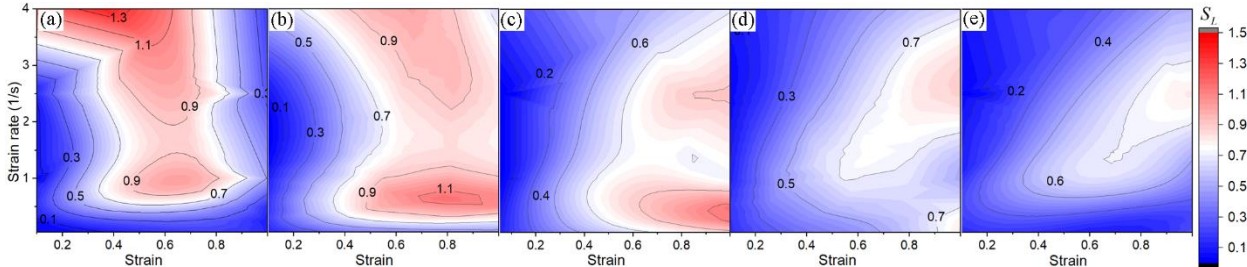

**Figure 16.** The effect of strain and strain rate on the density of low-angle boundaries $S_L$ depending on temperature, °C: (**a**) 300, (**b**) 350, (**c**) 400, (**d**) 450, and (**e**) 500. The dependences have been obtained from neural network predictions.

### 3.3. Analysis of Microstructure Formation Based on Neural Network Data

Figure 17 shows the dependences $D - \varepsilon$, $P_L - \varepsilon$, and $S_L - \varepsilon$ at a strain rate of 0.5, 2, and $4\,\text{s}^{-1}$ for temperatures between 300 and 500 °C. These dependences are based on the maps shown in Figures 14–16. The analysis of these dependences and maps of the microstructure parameters (Figures 14–16) yields the following generalizing conclusions.

1.  The strain dependences of the fraction of low-angle boundaries ($P_L - \varepsilon$) for the entire studied temperature and strain rate range of deformation have a peak (shown by asterisks in Figure 17).

2.  For the temperature range between 300 and 350 °C, the strain corresponding to the peak on the dependences decreases with an increase in strain rate. Conversely, for the temperature range between 450 and 500 °C, the strain corresponding to the peak increases with strain rate.

3.  For the temperature range between 300 and 500 °C and strain rates above $3\,\text{s}^{-1}$, the strain corresponding to the peak value of the dependence $P_L - \varepsilon$ increases with temperature (Figures 15 and 17).

4.  At low strain rates ($\dot{\varepsilon} < 1s^{-1}$), for the temperature range between 300 and 450 °C, the dependence $D - \varepsilon$ consists of two characteristic portions. In the first portion of the curve $D - \varepsilon$, the average grain diameter decreases, and it remains unchanged with increasing strain in the second portion (steady-state portion). Moreover, as the deformation temperature rises, the strain corresponding to the boundary between these two portions ($\varepsilon'$) shifts towards higher strains.

5.  For the temperature range between 350 and 500 °C, at strain rates exceeding $1\,\text{s}^{-1}$, the value of the average grain diameter decreases monotonically.

6.  An increase in the deformation temperature leads to the formation of a coarser-grained microstructure.

7.  The density of low-angle boundaries $S_L$ at the initial stage of deformation increases with strain for the entire range of temperatures and strain rates.

8.  At a temperature of 300 °C, after reaching a certain strain value (indicated by squares in Figure 17), the density of low-angle boundaries decreases with further deformation, as is the case with 350 °C, but at strain rates above $3\,\text{s}^{-1}$. For all the other deformation temperatures, the density of low-angle boundaries increases with strain or remains almost unchanged.

9.  At a temperature of 300 °C, the strain at which the density of low-angle boundaries $S_L$ decreases with increasing strain shifts towards lower strains with increasing strain rate.

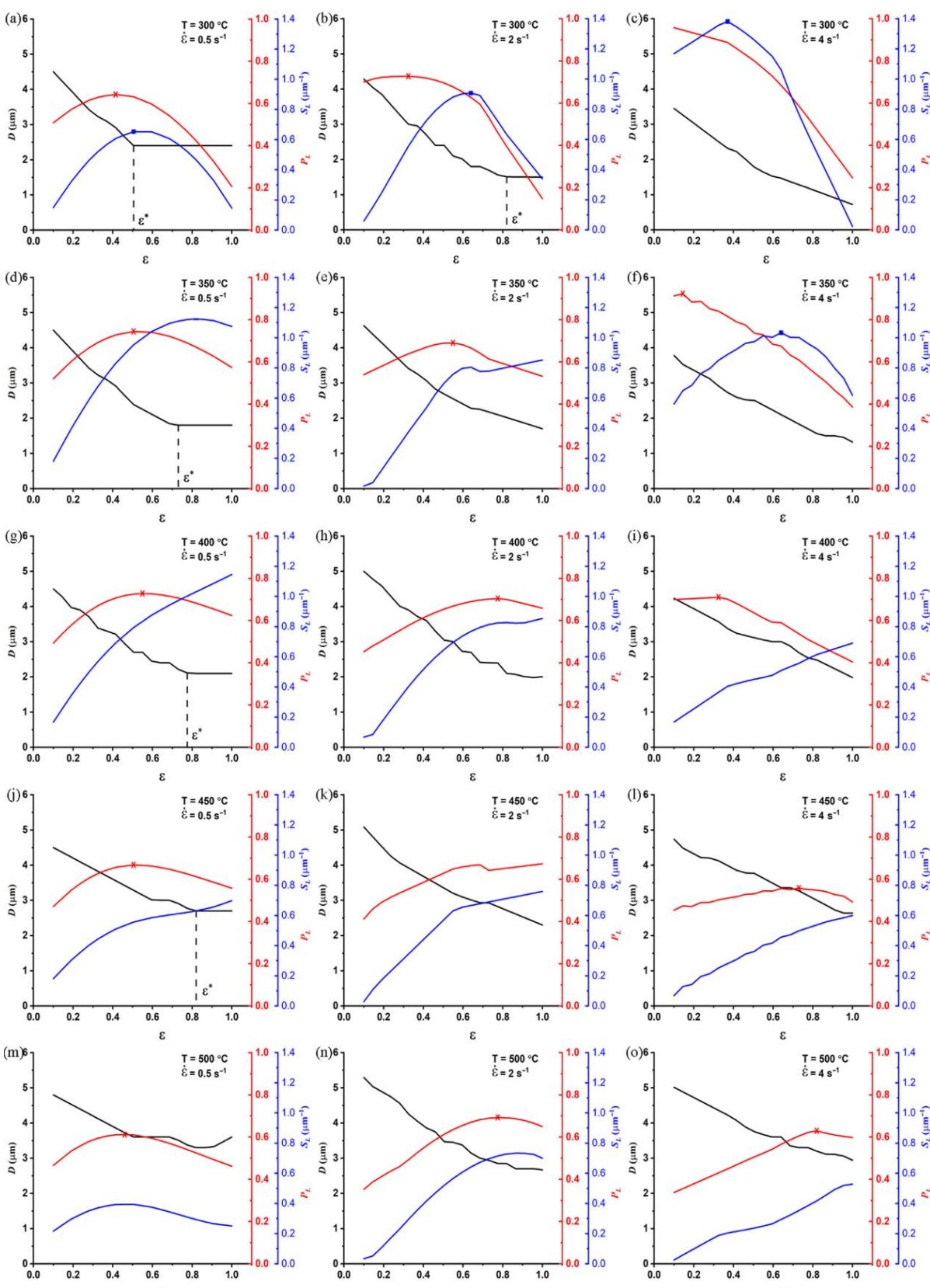

**Figure 17.** The average grain diameter $D$, and the fraction $P_L$ and density $S_L$ of low-angle boundaries as dependent on temperature $T$, strain $\varepsilon$, and strain rate $\dot{\varepsilon}$. The dependences are given for temperature, °C: (**a–c**) 300; (**d–f**) 350; (**g–i**) 400; (**j–l**) 450 and (**m–o**) 500.

The temperature, strain, and strain rate dependences of the fraction of low-angle boundaries $P_L$ for continuous dynamic recrystallization characterizes the influence of these thermomechanical parameters on the formation of new low-angle boundaries and their transformation into high-angle ones. At strains below the strain corresponding to the peak of the dependence $P_L - \varepsilon$, the rate of the formation of new low-angle boundaries prevails over the rate of the transformation of low-angle boundaries into high-angle ones. When the peak is passed, in the region of higher strains, the rate of the transformation of low-angle boundaries into high-angle ones prevails over the rate of the formation of low-angle boundaries. As can be seen from Figure 17, the dependence $P_L - \varepsilon$ of the composite matrix for the temperature and strain rate range under study is described by a convex upwards function. At the same time, under certain strain rate conditions, at temperatures of 300 and 350 °C, the strain dependence of the density of low-angle boundaries ($S_L - \varepsilon$) also has a peak, after which the fraction of low-angle boundaries decreases (Figure 17a–f). This indicates slower polygonization. Judging by the obtained microstructure images for these temperatures, the grains are close in size to the previously formed subgrains, and this indicates the transformation of subgrains into grains. At the same time, at a deformation temperature of 300 °C, the diameter of many subgrains at strains above 0.6 is close to the grain thickness (Figure 5). Together with the data on the evolution of the density and fraction of low-angle boundaries, this testifies that geometric recrystallization occurs in the composite matrix [43,67,68]. The composite matrix has a similar strain dependence of the fraction and density of low-angle boundaries at a temperature of 350 °C and strain rates above 3 s$^{-1}$. This gives grounds to assert that geometric recrystallization occurs in the composite matrix simultaneously with dynamic continuous recrystallization at a temperature of 350 °C, strains above 0.6 (Figures 15–17), and strain rates above 3 s$^{-1}$. Apparently, geometric recrystallization is not implemented at temperatures between 400 and 450 °C, strain rates between 0.1 and 4 s$^{-1}$, and strains below 1.

## 4. Conclusions

The rheological behavior and softening mechanisms in an AMg6/10% SiC metal matrix composite at temperatures from 300 to 500 °C and strain rates ranging between 0.1 and 4 s$^{-1}$ have been studied. It has been found from mechanical tests that, for the studied range of thermomechanical action, the flow stress curve has a peak value shifted towards higher strains with decreasing temperature and increasing strain rate in the entire range of temperatures and strain rates. This rheological behavior of the composite is due to the interaction of hardening and softening processes. In order to study the mechanisms of softening occurring during deformation, neural networks were built, which required the development of a new technique for processing experimental data, which allows one to form a training sample necessary for a correct description of the initial experimental data by the neural networks. The constructed neural networks have made it possible to describe the evolution of the average grain diameter, and the fraction and density of low-angle boundaries depending on strain, strain rate, and deformation temperature with acceptable accuracy. The use of neural networks together with EBSD images of the formed microstructure during deformation has allowed us to identify the occurrence of the following relaxation mechanisms depending on the thermomechanical conditions of deformation:

1. At a temperature of 300 °C and strain rates ranging between 0.1 and 4 s$^{-1}$, the composite matrix softens by dynamic recovery and continuous recrystallization, and when a certain value of strain is reached, geometric recrystallization occurs in some grains. At the same time, the strain at which geometric recrystallization starts to intensify in the grains shifts towards lower strains with an increasing strain rate.
2. At a temperature of 350 °C, geometric recrystallization, together with continuous recrystallization, occurs in the composite matrix at strain rates above 3 s$^{-1}$.
3. At temperatures from 400 to 500 °C and strain rates ranging between 0.1 and 4 s$^{-1}$, the main softening processes are dynamic recovery and continuous dynamic recrystallization.

**Author Contributions:** Conceptualization, A.S.; methodology, A.S. and V.K.; validation, A.S. and V.K.; formal analysis, A.S. and V.K.; investigation, A.S. and V.K.; writing—original draft preparation, A.S. and V.K.; writing—review and editing, A.S. and V.K.; project administration, A.S. and A.K.; funding acquisition, A.S. All authors have read and agreed to the published version of the manuscript.

**Funding:** This research was funded by the RSF (project No. 22-29-00428) in the part of constructing models of metal matrix composites.

**Institutional Review Board Statement:** Not applicable.

**Informed Consent Statement:** Not applicable.

**Data Availability Statement:** The data presented in this study are available on request from the corresponding author.

**Acknowledgments:** The Plastometriya shared the research facilities of the IES UB RAS, and the shared research facilities of the B. N. Yeltsin Ural Federal University were used. The work on studying the rheological properties of the composites was performed within the research conducted by the Institute of Engineering Science, Ural Branch of the Russian Academy of Sciences, project No. AAAA-A18-118020790140-5.

**Conflicts of Interest:** The authors declare that they have no conflict of interest. The funders had no role in the design of the study; in the collection, analyses, or interpretation of data; in the writing of the manuscript, or in the decision to publish the results.

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
