# Peer review of "Neural Network Modeling of Microstructure Formation in an AlMg6/10% SiC Metal Matrix Composite and Identification of Its Softening Mechanisms under High-Temperature Deformation"

_applsci, doi:10.3390/app13020939_

Round 1

Reviewer 1 Report

I have read the article titled" The rheological behavior and microstructuring of the AlMg6/10% SiC metal matrix composite at high-temperature deformation" for possible publication in the Applied Sciences journal. I am recommending a minor revision for this submission. My comments are attached herewith.

Please change to " The rheological behavior and microstructure of AlMg6/10% SiC metal matrix composite at high-temperature deformation"

Author Response

The authors of the article thank the reviewer for reading the article. The responses to the comments are given below.

Q1. Introduction needs to be checked again. what is the novelty of the study? what is the difference between present study and previous studies? what are the objectives? research gap. all these should be addresed in the introduction. I would suggest the authors to read the below article and cite in the literature

Ganesh, M. R. S., Reghunath, N., J.Levin, M., Prasad, A., Doondi, S., & Shankar, K. V. (2022). Strontium in Al–Si–Mg alloy: A review. Metals and Materials International, 28(1) doi:10.1007/s12540-021-01054-y

We corrected the introduction and title of the article.

Q2. What is the purpose of comparing the alloys and composites?

The alloys in the table are alloys that are similar in application to the AlMg6/10% SiC composite, that is, they can be used in the same structures. The properties are given just for comparison.

Q3. How was this parametr fixed ?

The temperature was controlled by a built-in thermocouple in a furnace. The annealing time was controlled by ourselves.

Reviewer 2 Report

The paper deals with the rheology of a kind of MMC as well as investigating the microstructural properties. The manuscript is well-structured and presents a detailed analysis. I think the study is useful for the field. However, an attempt is needed to publish it in the journal. Here the comments.

·        Please check the text for typos.

·        Please clearly indicate the novelty in this work.

·        Is there a preliminary study about size of SiC particles? How did authors decide on the particle size?

·        I think the chemical composition should be given in a Table instead of giving in the text.

·        Metallurgical sample preparation stages should be given in the text.

·        What should be understood from Fig.1? Please discuss this figure. Color indicators should be given with Fig.1.

·        How did authors obtain the materials data as the input of FEA program?

·        Similar to the previous comment, how did authors decide on the friction as 0.09? Authors should support them with experimental data or early studies in the literature.

·        EBSD images are in a mess. Please reorganize them to present in a better way in figures.

·        Comparing the MMC and Al alloys with no fillers at the same conditions, what is the role of SiC content in the material? Is there any other microstructural evolution or a different response of alloy with no SiC loading?

·        Please give an equation number for the formulas.

·        Conclusions should be rewritten by discussing the result instead of using bullet points.  

Author Response

The authors of the article thank the reviewer for reading the article. The responses to the comments are given below.

  • Please check the text for typos.

Thank you for your careful reading. Typos have been corrected.

  • Please clearly indicate the novelty in this work.

We corrected the introduction and title of the article.

  • Is there a preliminary study about size of SiC particles? How did authors decide on the particle size?

The granulometric composition of the powders was determined using a LaSca-TD.

This information was included in the article text.

  • I think the chemical composition should be given in a Table instead of giving in the text.

We made changes in the article text.

  • Metallurgical sample preparation stages should be given in the text.

The initial components of the AlMg6/10% SiC MMC were mixed in a vibratory mixer in an argon atmosphere. Sintering was carried out for 60 minutes at a temperature of 420 °C. The pressure at which sintering occurred was 30 MPa. No additives modifying the surface of SiC particles were used.

Information about the preparation of the composite was added in the article text.

  • What should be understood from Fig.1? Please discuss this figure. Color indicators should be given with Fig.1..

Discussion was added to the article text. Figure 1 was corrected.

  • How did authors obtain the materials data as the input of FEA program?

Compression experiments were carried out by an automated plastometric installation designed at the Institute of Engineering Science, UB RAS.

This information was included in the article text.

  • Similar to the previous comment, how did authors decide on the friction as 0.09? Authors should support them with experimental data or early studies in the literature.

Previously, we did tests for determining the friction coefficient of the lubricant for aluminum alloys. We conducted it for the AlMg6 alloy. In these experiments, we deformed specimen at different heights. We selected the friction coefficient in the Coulomb friction law so that the maximum Dmax and minimum Dmin diameter of the specimens coincided as closely as possible with the calculated one at different strains. The explanatory figure is given in the attachment. We added a reference to the technique in the article text.

  • EBSD images are in a mess. Please reorganize them to present in a better way in figures.

The EBSD images are sorted by increasing the strain. The apparent randomness is due to the fact that the second parameter (the strain rate) in this case turns out to be chaotic. Unfortunately, there is no other way.

  • Comparing the MMC and Al alloys with no fillers at the same conditions, what is the role of SiC content in the material? Is there any other microstructural evolution or a different response of alloy with no SiC loading?

No studies have been done for the AlMg6/10%SiC. For a 7075/10%SiC and a 7075/0%SiC made by powder technology, we have conducted such studies. The addition of SiC particles to the powder of the 7075 alloy resulted in the formation of a finer-grained microstructure than in 7075/0%SiC. The rate of relaxation mechanisms also changed, which led to different dependences of the fraction of low-angle boundaries on temperature, strain rate and strain.

  • Please give an equation number for the formulas.

We added equations numbers.

  • Conclusions should be rewritten by discussing the result instead of using bullet points.

We corrected the conclusion.

Round 2

Reviewer 2 Report

The revised version is acceptable. 

Author Response

The authors are grateful for the help in improving the article.